# A “Numerical Evo-Devo” Synthesis for the Identification of Pattern-Forming Factors

**DOI:** 10.3390/cells9081840

**Published:** 2020-08-05

**Authors:** Richard Bailleul, Marie Manceau, Jonathan Touboul

**Affiliations:** 1Developmental Biology & Cell Biology and Biophysics Units, European Molecular Biology Laboratory, Meyerhofstraße 1, 69117 Heidelberg, Germany; richard.bailleul@embl.de; 2Centre for Interdisciplinary Research in Biology, CNRS UMR 7241, INSERM U1050, Collège de France, 75005 Paris, France; 3Department of Mathematics and Volen National Center for Complex Systems, Brandeis University, 415 South Street, Waltham, MA 02453, USA

**Keywords:** pattern formation, natural variation, modelling

## Abstract

Animals display extensive diversity in motifs adorning their coat, yet these patterns have reproducible orientation and periodicity within species or groups. Morphological variation has been traditionally used to dissect the genetic basis of evolutionary change, while pattern conservation and stability in both mathematical and organismal models has served to identify core developmental events. Two patterning theories, namely instruction and self-organisation, emerged from this work. Combined, they provide an appealing explanation for how natural patterns form and evolve, but in vivo factors underlying these mechanisms remain elusive. By bridging developmental biology and mathematics, novel frameworks recently allowed breakthroughs in our understanding of pattern establishment, unveiling how patterning strategies combine in space and time, or the importance of tissue morphogenesis in generating positional information. Adding results from surveys of natural variation to these empirical-modelling dialogues improves model inference, analysis, and in vivo testing. In this evo-devo-numerical synthesis, mathematical models have to reproduce not only given stable patterns but also the dynamics of their emergence, and the extent of inter-species variation in these dynamics through minimal parameter change. This integrative approach can help in disentangling molecular, cellular and mechanical interaction during pattern establishment.

## 1. Introduction

Natural patterns have been classified in symmetries, trees, fractals, spirals, meanders, waves, foams, tessellations, cracks, stripes, and dotted arrays [1], and strikingly, examples for most can be found in animals. A long-standing scientific challenge has been to unravel the chain of developmental steps through which these intricate motifs emerge, from first symmetry-breaking events in initially homogeneous structures to the timely propagation of pattern-forming competence and the differentiation and spatial arrangement of patterned characters in typical geometries. Uncovering these processes implies bridging phenomena occurring at the molecular, cellular, tissue, organismal, and population levels, and together, developmental genetics, evolutionary biology and mathematics paved the way to identifying pattern-forming factors in biological systems, respectively taking advantage of model organisms, natural variation, and modelling. We discuss here how empirical and theoretical approaches led to the formulation of two major patterning theories, and review recent studies in model and non-model vertebrates, showing that combining these approaches sets a foundation for powerful work in pattern-forming studies. 

## 2. Diversity vs. Stability of Natural Patterns: A Paradox Guiding Methods and Model Choices

The striking diversity of motifs created by the spatial arrangement of appendages, pigments, segments, etc., partly results from the complex combination and superimposition of many patterns across the body. It has been associated to the multiplicity of physiological and adaptive functions at the basis of evolutionary change [2,3,4,5,6,7,8,9]. However, within species or taxa, most patterns display similar periodicity (i.e., number of repetitions within a period of space or time) or orientation along body axes, a stability thought to guarantee survival and reproductive success in a given niche. This apparent paradox raised enormous interest from developmental biologists, evolutionary biologists and mathematicians, with pattern conservation being rooted in the spatio-temporal hierarchy of pattern-forming events during embryonic development [10], and the extent of pattern variation reflecting developmental constraints to evolution. Researchers from these different fields tackled pattern formation using methodologies based on an opportunistic choice of study systems. Developmental studies have taken advantage of easily tractable designs present in model systems and sharply defined in space and time. Emblematic examples include the antero-posterior distribution of segments and the spatial arrangement of bristles, wing veins, and ommatidia in drosophila (e.g., [11,12,13]), or the production of skeletal structures, teeth circumvolutions, appendage arrays, or colour patterns in fish, birds or rodents (e.g., [14,15,16,17]). The availability of genetic tools and the possibility to perform functional experiments in model organisms enabled functionally dissecting core developmental mechanisms and linked those to the production of conserved attributes of natural patterns. In particular, morphogens, molecules that diffuse from developing sources and form local concentration gradients, have been repeatedly implicated in pattern formation ([18] and see below). However, their expression profiles do not necessarily correlate spatially with patterns, complicating the selection of candidates. In addition, analysing pattern phenotypes through functional tests or genetic screens rarely allows disentangling mechanisms necessary for character production from those involved in their spatial distribution. Using groundwork from developmental genetics, many “evo-devo” studies recently focused on pinpointing the molecular or tissue basis of visible differences observed between homologous characters of non-model organisms. Quantitative genetics and comprehensive comparative expression analyses have been performed in a wide range of invertebrates, from ladybugs to butterflies or nematods [19,20,21], and vertebrates such as stickleback, cichlid, or cave fish, and African striped or deer mice, etc. [22,23,24,25]. This work provided insights into the genetic basis and late acting cellular/tissue events governing evolutionary changes in the orientation, repetition or geometry of patterned characters [22,23,26]. Because of technical challenges in natural populations, evo-devo studies, however, face the general difficulty of functional validation, and are often limited to large-scale correlations or to a few varying species. Finally, mathematical modelling has been central to pattern formation studies. Natural designs are spatial combinations of basic units, arranged so as to optimize steric space [27] or accommodate physical constraints [28], and corresponding to mathematically describable geometries: circles, straight/curved lines, squares, or polygons. A few mathematical concepts may thus describe numerous biological observations. Partial differential equations (PDEs) provide a useful framework, since by nature they create spatio-temporal dynamics. Their simulations in silico reproduce changes in tissue morphology, molecular concentrations or cell motility, naturally generating stable patterns of basic units as observed in vivo. Modelling PDEs thus enabled predicting biological conditions necessary to pattern formation such as the action and interaction of molecular and cellular factors ([29,30] and see below). In theoretical fields, choosing models also imposes limitations: several mathematical models with different behaviours, potentially representing different biological processes, can accurately reproduce a single pattern. Conversely, a single model can often generate a variety of patterns depending on parameters. In addition, the logic of using in silico simulations is often used to reproduce final pattern states but rarely the developmental paths to their formation. In sum, biological work identified putative patterning factors, but rarely allowed in depth mechanistic understanding of their mode of action. Conversely, patterning mechanisms were theoretically predicted, but deriving prediction for precise biological insights has been difficult to extract from modelling. Research fields and their inherent choices of models inevitably restricted and biased technical and conceptual approaches, which led to different views on the nature and mode of action of patterning factors. We present below the two main patterning theories formulated to date.

## 3. Two Long-Opposed Theories Explain Pattern Formation In Vivo

Alternative patterning dynamics have been proposed to explain the emergence of patterns. In instructional patterning, cells adopt fates according to the amounts of positional information acquired from an external source. Lewis Wolpert conceptualized in 1969 the “French flag model”, in which morphogen gradients and differentiation thresholds create distinct compartments in a developing tissue. Biological evidence ensued: arguably the most emblematic is the periodic expression of *Gap* genes providing antero-posterior identity to segments in the drosophila embryo, governed by gradients of *Bicoid* maternal mRNA [31]. Morphogens have since been involved in invertebrate and vertebrate patterning and shown to possess properties compatible with instructional signalling [32,33,34]). A major strength of this theory is that it intuitively explains the orientation of many periodic patterns along body axes: pattern directionality would be given by candidate positional signals emanating from early axial structures (e.g., neural tube, somite). However, instructive signalling hardly reconciles with the diversity and complexity of patterns, nor does it provide mechanistic understanding of some attributes such as periodicity. In self-organisation, intrinsic instabilities within the initially homogeneous tissue spontaneously cause its arrangement in a pattern. Alan Turing first formulated this theory in 1952 with a “reaction-diffusion” model describing the interaction of an activator and long-range diffusing inhibitor. Most work has since assumed a molecular basis for self-organisation. Morphogens, in particular, diffuse at long range and have expression levels or degradation rates compatible with self-organising dynamics [34]. Self-organising models have been repeatedly used to recreate animal designs in numerical simulations. Their instabilities have inherently stochastic parameters (e.g., diffusion, attraction) with excitable or oscillatory behaviours able to generate repetition, and universal properties such that a few simple conditions, related to the model’s equations evaluated at the homogeneous state, can lead to symmetry breaking and pattern emergence. In addition to being efficient at producing periodic patterns, self-organisation also provides an appealing explanation for how natural variation arises, sometimes rapidly at an evolutionary scale, as minimal parameter variation often results in important pattern differences in simulations (e.g., [35]). The malleability of self-organising dynamics, however, does not account for the pattern reproducibility and directionality seen in nature. 

Long opposed, instructive signalling and self-organisation are now both viewed as major patterning processes, likely combining in space and time to form many patterns [36]. This synthesis was made possible through the combination of numerical and empirical work, which integrated molecular, cellular, and mechanical components of tissue development. For example, Turing models can recover the longitudinal orientation of fish coloured stripes in silico when simulated in frames seeded with non-homogeneous axial initial conditions [37], or when modulated by production/degradation gradients or tissue anisotropy [38], all providing instructive information. Similarly, combining PDEs describing competition for gene expression, auto-catalytic activity and small diffusion of molecular factors with positional information provided by transcription factors recreates in silico the regular boundaries observed between brain areas in the developing nervous system [39,40]. Biological work also partly reconciled the two patterning strategies. Teeth spatially arrange through Shh-dependent self-organising events arising from so-called “signalling centers” acting as instructive sources [41], and the patterning waves that characterise the timely segmentation of the vertebrate mesoderm in somites are triggered by early instruction and once propagating, generate periodic patterns de novo (e.g., [37,42]). In juvenile poultry birds, longitudinal bands of agouti expression foreshadow periodic coloured stripes adorning the dorsum: their width relies on a late control of agouti’s dose by pigment cell self-organisation, while their absolute position is controlled by early instructive signals from the somite [43]. Taken together, results show that pattern formation combines molecular induction, cell/tissue autonomous processes such as self-organisation, and a morphogenetic response to positional cues (i.e., the integration of positional information by cells creating tissue shape changes). For given organs displaying motifs, these processes repeat over time and at different spatial resolutions, in a progressive refinement of space. A current challenge is thus to understand how patterning strategies emerge and combine in space and time to create robust designs in vivo.

## 4. Novel Integrative Frameworks Bridge Molecular and Cellular Patterning Strategies

In this endeavour, recent inter-disciplinary dialogues combined developmental biology with comprehensive numerical approaches, increasing predictive power and allowing better design of empirical tests, thereby providing novel insights into how patterning strategies combine. We mention here four study systems recently used to implement such approach in mice and birds. A bulk of work first took advantage of digit patterning during limb growth in mice. This process involves dynamic interaction between molecular factors, including Bmp2, Sox9, and Wnt, whose expression form striped patterns, and Hox and Fgf, which form gradients from a distal source. A mathematical model comprising Turing-like reaction-diffusion, where the molecular species are diffusive Bmp2 and Wnt and non-diffusive Sox9, external cues describing Hox and Fgf gradients [44] and realistic tissue growth [45], reproduced proper dynamics of Sox9 expression. Here, and consistent with the recent synthesis of patterning theories presented above, self-organisation is combined with instructive gradients to control the wavelength of the striped pattern [46]. Sadier and colleagues studied the formation of rodent teeth, which emerge through complex timely dynamics prior to stabilization in a final pattern. Sequential *Edar* expression in signalling centres was used to build a mathematical model composed of Turing-type dynamics (short range activation and long-range inhibition triggered by asymmetric bi-stable regime) and cell density and movement towards the activator through chemotaxis [47,48]. Simulations were run on growing domains corresponding to the antero-posterior axis of the dental epithelium. This reproduced the sequence of tooth centre emergence, predicting that newly formed structures can “erase” previously formed ones, a hypothesis that was then functionally validated in vivo. In both studies, modelling was used to recreate observed dynamics of molecular expression and character production rather than only a final pattern. Predictions thereby suggested that simple rules govern apparently complex spatio-temporal regulation during tissue development, allowing authors to explain the outcomes of experimental inhibition of Bmp or Wnt [49] or functional effects of *Edar* loss-of-function [41]. 

Similar approaches were used to study the timely establishment of feather follicle arrays in avian skin. In poultry birds, feather follicles arrange in a hexagonal array (each follicle is surrounded by six neighbours), and differentiate through dynamic molecular-cellular interaction: FGF signals from the epidermis cause local aggregation of mesenchymal cells, which activates BMP signals, and in turn, further FGF production in the epidermis. By integrating reaction-diffusion akin to Turing’s model and chemotaxis, mathematical modelling predicted this interplay, providing self-organising conditions sufficient to break symmetry and create a periodic dotted pattern visually resembling the nascent follicle array [50]. Ho and colleagues then applied to this model a mathematical wave fitting simulation to the dynamic emergence of the follicle array in the domestic chicken, in which previous morphological description showed that follicles individualise in longitudinal rows through a medial-to-lateral wave of differentiation occurring within “tracts”, which are areas of the skin which are able to form feathers [27]. Using this prediction as head-start, they searched for in vivo markers of a wave, and showed that both cell density and *Eda* expression increase progressively, travelling laterally prior to follicle formation. This suggested that follicle-forming competence is attained when cell density reaches a threshold whose value depends on the presence of *Eda* signalling. In this study, empirical work was built on modelling predictions to identify molecular and cellular dynamics characterising a morphogenetic wave [51]. Taking advantage of the same bulk of theoretical work, our laboratory studied how the follicle formation wave is initiated and progresses in a timely fashion. We first worked out to recreate the follicle array and dynamics of its emergence without resorting to an extrinsic mathematical wave. We combined the two preeminent processes previously used, namely reaction-diffusion and chemotaxis, with a third mathematical term explicitly describing cell proliferation as an autonomous logistic function. This term, inspired from modelling studies of virus spreading and population behaviour, implements growth in systems with resource limitations [52,53], and classical theories of diffusion equations with logistic growth are known to develop traveling waves, spreading initial perturbations through directional nearest-neighbour contagion [29,30]. The unified model of three partial differential equations applied to simulations frames seeded with axial initial conditions intrinsically reproduces sequential dynamics and the directionality of follicle array formation. This observation produced the first predictions: coupling self-organisation with proliferation triggers timely emergence of follicles; adding initial conditions provides directionality, together creating a patterning wave. Analysing and testing parameters of the model provided a second prediction: while self-organisation controls follicle individualisation, as previously shown, the proliferation rate of the logistic source controls the duration of the patterning process. We investigated the relevance of this model-based prediction by analysing dynamics of cell proliferation throughout tract differentiation in the Japanese quail, a species closely related to the domestic chicken, and confirmed the existence of a travelling front of increased cell density that progressively provides follicle-forming competence to longitudinal domains as it propagates laterally. Strikingly, the ratio of proliferative cells linearly decreased with respect to cell density, ceasing when the skin tissue attains a certain carrying capacity (i.e., a cell density threshold), showing that cell proliferation occurs according to predictions of a logistic growth term—a unique in vivo validation of this widely used mathematical equation. We used drug treatments on skin explants, inhibiting proliferation without altering the carrying capacity of the skin, to demonstrate that proliferation rate governs pattern duration, and also supporting the predictions of the model. In our study, accuracy of modelling was necessary to help design empirical experiments. Together, these three studies uncover similarities between patterning mechanisms in different systems (e.g., the important roles of morphogen and Eda/EdaR signalling, cell density/proliferation, and self-organisation; Figure 1). They illustrate the strength of establishing numerical-empirical crosstalk in pattern-forming studies. We argue below that adding elements of natural variation to in silico work is key to improving this crosstalk. 

## 5. A New Synthesis: Integrating Natural Variation to Numerical-Empirical Crosstalk

Ho and colleagues extended observations to *Paleognathae*, an ancestral group comprising flightless birds. They found that both Eda signalling and the follicle patterning wave are absent in the ostrich, which forms follicles simultaneously within tracts, and in the emu, which forms follicles with a delay, with earlier lower cell density skin impeding competence for pattern formation. The *Paleognathae* follicle pattern is irregular, and authors thus proposed that the progressive acquisition of competence ensures hexagonal fidelity to the follicle pattern in domestic chickens. This work exemplifies the value of taking into account natural variation in a given pattern, a concept we extended and systematized ahead of modelling: in addition to final patterns and their dynamics of formation, the objective of our unified model was to accurately anticipate inter-species differences in this process. We performed a comparative survey by marking nascent feather follicles with *β-catenin* in the emu, representing *Paleognathae*, the domestic chicken, the Japanese quail, and their relative in the *Galleoanserae* group, the common pheasant, and two members of the third, species-rich group *Neoaves*, namely the zebra finch, passerine songbird, and a species of emblematic penguins, devoid of flight abilities. We observed variation in their formed follicle pattern, as well as in the location and shape of early *β-catenin* expression in the un-patterned tract: *β-catenin* marked segments of variable shapes in all poultry birds and the zebra finch, and larger bilateral areas in emus and penguins. In addition, the follicle array emerged in a row-by-row sequence in the first group, and apparently simultaneously in emus and penguins. This observation was consistent with the proposed link between a loss of flight abilities and of the patterning wave, but as we found the penguin pattern to be extremely regular, pattern regularity may be controlled by other means. Surveying a large number of species allowed for correlating differences in initial *β-catenin* expression to pattern sequentiality: we thus ran simulations of our model on simulation frames seeded with species-specific initial conditions. This recreated the species-specific dynamics of pattern emergence, producing a third prediction: initial conditions control wave sequentiality; their variation between birds is responsible for differences in the timing of the follicle patterning process (Figure 2). We thus varied parameters of initial conditions and showed that row-by-row dynamics occur only when initial conditions are spatially restricted and sharp enough. Further work is now necessary to identify in vivo factors displaying such expression profiles and test their role in the sequential control of wave initiation and/or progression. Appealing possibilities include molecules emanating from neighbouring neural tube or somites, structures that differentiate longitudinally and instruct the formation of epidermal and dermal derivatives, including the striped colour pattern adorning the dorsum of poultry birds [44,54,55]. In this study, tracking various developmental paths to pattern establishment and using these as basis for modelling increased the precision of simulations and better guided empirical work. In addition to generating hypotheses with key implication in evolutionary biology and ecology (e.g., here, on the loss of flight abilities), natural variation is thus a powerful tool, raising the bar in the extent to which biological data can be used for model building, and conversely, in the accuracy of numerical simulations. 

## 6. Designing Numerical Evo-Devo Approaches to Study Tissue Mechanics

The unified models described above are not exhaustive, and do not account for a number of other events experimentally shown to contribute to pattern formation. Tissue growth, for instance, was modelled in studies of mouse digit and teeth patterns [41,48,56,57] or periodic stripes in zebra fish tailfins [58,59], but reciprocal interactions between molecular, cellular, and mechanical mechanisms remain absent from most models. For example, all known cellular and molecular markers of follicle differentiation, from cell proliferation to the fluctuation of molecular expression, may be generated or affected by intrinsic mechanical properties of the cutaneous tissue. The onset of follicle emergence is indeed characterised by the nuclear translocation of β-catenin in epidermal cells, an event suggested to be triggered by contractile properties of aggregating dermal cells [60]. Similarly, epidermis compression has been shown to intensify FGF expression [51]. Tissue properties consequentially shape the resulting patterns. For example, modifying dermal cell contractility or substrate stiffness ex vivo alters the size and spacing of follicles [60]. More generally, in several other systems, the skin physical forces exerted on/by cells by/on the extra-cellular environment (e.g., shear stress, compression, tension, traction, adhesion) have been linked to changes in extra-cellular matrix architecture, cell cycle, cell motility and signalling [61,62,63,64,65,66], likely playing a defining role in patterning processes. With the advent of biophysical tools to measure physical parameters in vivo [67] and the ever growing amount of theoretical frameworks integrating biomechanics [68,69,70,71], it now becomes possible to explore the role of tissue mechanics with comprehensive experimental modelling approaches. One may infer previous unified models by adding explicit dependence of some parameters of reaction-diffusion and chemotaxis terms on mechanical parameters (e.g., molecular diffusion could be a function of substrate stiffness [72]). A more direct approach would be to use purely mechanical models, previously shown to produce periodic designs such as dotted arrays in one or two dimensional spaces [73,74]. Alternatively, mechanical and chemical couplings proved powerful: necessary Turing model’s conditions were obtained by coupling biomechanical parameters promoting short-range activation and long-range inhibition to only one morphogen, suggesting a crucial role for tissue mechanics in pattern formation [69,75], and providing a new paradigm relaxing the necessity of an inhibitor diffusing at a long range—an important constraint of Turing’s model which is yet to be supported in many in vivo systems [76]. In light of the studies described above, numerical methodologies integrating molecular, cellular and mechanical elements may greatly benefit from taking into account the developmental dynamics of pattern formation, and adding elements of variation in the architecture and bio-physical properties of the developing tissue as observed between species displaying relevant pattern differences. Such approaches will pave the way to identifying the relative implications of mechanical and chemical events in pattern formation and evolution.

## 7. Conclusions

The recent literature shows that biological and computational research fields can establish mutually beneficial crosstalk to uncover mechanisms at play in pattern formation. The extensive knowledge acquired in developmental genetics and increasing feasibility of in vivo tests, thanks to technical advances in molecular, imaging and bio-physical tools, better defines biological parameters in mathematical models, whose simulations increase in resemblance to patterns and patterning dynamics observed in living organisms. Conversely, the unification of partial differential equations and progresses in model analyses provide predictions which are less prone to ambiguity to guide empirical tests. When theory is guided by alterations shaped by evolution, including results from now-feasible functional experimentation in non-model organisms [77,78], or data from surveys of natural variation [37], it allows for shedding many pattern-forming dynamics to their greatest simplicity. Such “numerical evo-devo” synthesis is the promise of novel insights in pattern-forming studies in the very near future.

## Figures and Tables

**Figure 1 cells-09-01840-f001:**
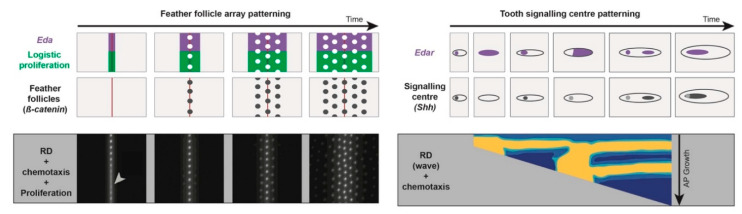
Similarities in molecular and cellular bases of patterning in different systems. During the patterning of both feather follicle arrays (marked by *β-catenin* expression) or tooth signalling centres (marked by *Shh* expression), *Eda/EdaR* allow visualizing morphogenetic waves, which are also characterised by a local increase in cell density and proliferation. In the first case, the wave travels along the medio-lateral axis, while in the second, timely regulation occurs during antero-posterior growth. Simulations of mathematical models composed of Turing-type reaction-diffusion, chemotaxis (and logistic proliferation in the case of follicles) can reproduce spatial dynamics of *EdaR*, *Eda*, or *β-catenin* expression and proliferation during tissue differentiation (high levels of *β-catenin* or *EdaR* expression respectively appear in white and yellow in simulations). Adapted from [37,41].

**Figure 2 cells-09-01840-f002:**
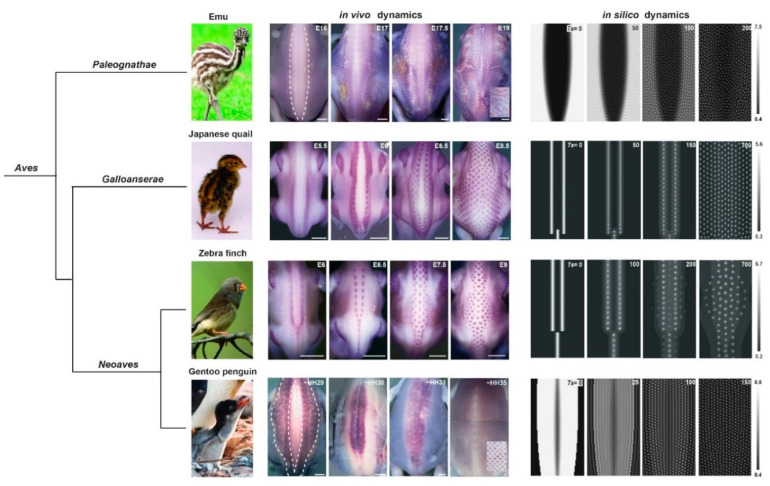
In vivo/in silico dynamics of dorsal feather follicle patterning in avian species. In the emu embryo, feather follicles appear simultaneously within large *β-catenin*-expressing areas initially covering the dorsal skin region expect for its medial-most part. In the Japanese quail and the zebra finch, dorsal follicles appear in a sequential manner from *β-catenin*-expressing longitudinal lines. In the Gentoo penguin, *β–catenin* is expressed in two bilateral expressing areas followed by a central area; follicles rapidly appear in the herein defined zones. Computer simulations of a unified model combining reaction-diffusion, chemotaxis and logistic proliferation recapitulate all dynamics when applied to species-specific axial initial conditions. This predicted that patterning sequentiality is due to spatially confined symmetry breaking in the dorsal skin of bird embryos. Adapted from [37]. Photo credits: Manceau laboratory, Paris, France (Japanese quail), Wikimedia (finch), Jooin (emu) and ©Raphaël Sané (www.raphaelsane.com; Gentoo penguin).

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
