# Peer review of "A “Numerical Evo-Devo” Synthesis for the Identification of Pattern-Forming Factors"

_cells, 2020, doi:10.3390/cells9081840_

Round 1
Reviewer 1 Report
Dear editor,
Dear authors,
I had to evaluate the review manuscript by R. Bailleul and colleagues entitled “A “numerical evo-devo” synthesis for the identification of pattern forming factors” submitted to Cells.
In this pleasant to read manuscript, the authors have managed to synthetized a large part of the knowledge related to pattern formation. Usually, this kind of collection is based on a specific model, but here the authors are covering various models and organs.
The review is really interesting as it compares different models around the same aspect of mathematical modeling of pattern formation.
While the work is already extensive and well presented, I feel that some works are missing here. Especially the studies related to cell behavior, which is the basis of pattern formation.
For instance, the work from Jernvall laboratory on tooth, or from Dhouailly laboratory on feather, both describing the mechanistic of pattern formation at cellular level, and descriptive mathematical models, is a missing part in this review to make it a perfect collection explaining all about pattern formation.
I really hope to see a modified version of this work covering these aspects of pattern formation.
descriptive and functional manuscript, the authors have used classical and up to date methods to understand the molecular mechanisms involved in successional murine molar formation.
In my opinion, this work is not only of interest for developmental biologists, but as well of interest for cellular, molecular and modelling experts. Hence, it will be a great addition to Cells, if the proposed modifications are applied.
Best wishes,
Author Response
We thank the reviewer for his comments on our manuscript. We have added description of work by J.Sharpe's laboratory on digit patterning however we feel that adding reference to work on tooth/feather follicle formation is out of the scope of this review. Here, we focus on how organs or appendages are spatially arranged, rather than signalling and morphogenetic events governing their differentiation.
Organogenesis work provides basis for identifying molecular and cellular candidates for a role in pattern formation. However the aim of this review is to discuss novel means to predict candidates and design experimental paradigms in pattern forming studies beyond organ differentiation, process that shall be disentangled from pattern formation. We thus limited references to the formation of organs/characters in the patterning systems we chose to decribe; those references can however be found in reviews we cite (e.g., Neguer et al., which includes reference to work from Dhouailly or Jernvall labs).
Reviewer 2 Report
The authors have done a great job to summarize how self-organization and instructive signals can control biological pattern formation during evolution. In the introduction and in the second section, the authors build the conceptual framework to understand the work presented in the third and fourth section. The main idea presented here is that instructive patterning and self-organization can act together to make self-organized patterning more "controllable" and robust. The authors cite most of the literature on the topic, but forgot to discuss some relevant studies:
a) In the second section they should discuss highly related work on digit patterning: see for example the study (Sheth et al. Science 2012) and the perspective (Muira T. Science sig. 2013), that talks about the same concepts. Also they should cite the limb development model developed in (Raspopovic et al. 2014) that combines tissue growth, morphogen gradients and a Turing model to reproduce the spatiotemporal dynamics of digit patterning. This work is highly relevant for the discussion presented in the third section.
b) In the second section, when they talk about the orientation of Zebrafish stripes they should cite the theoretical study presented in (Hiscock and Megason 2016). Moreover, they should cite the work of Alxeandria Volkening (Volkening 2020 and Volkening et al 2020), which is one of the best example of integrative modeling of self-organization at different levels.
Sheth R, Marcon L, Bastida MF, Junco M, Quintana L, Dahn R, Kmita M, Sharpe J, Ros MA (2012). Hox genes regulate digit patterning by controlling the wavelength of a Turing-type mechanism.
Miura T. Turing and Wolpert work together during limb development. Sci Signal. 2013;6(270):pe14. Published 2013 Apr 9. doi:10.1126/scisignal.2004038
Hiscock TW, Megason SG. Orientation of Turing-like Patterns by Morphogen Gradients and Tissue Anisotropies. Cell Syst. 2015;1(6):408-416. doi:10.1016/j.cels.2015.12.001
Volkening A. Linking genotype, cell behavior, and phenotype: multidisciplinary perspectives with a basis in zebrafish patterns [published online ahead of print, 2020 Jun 27]. Curr Opin Genet Dev. 2020;63:78-85. doi:10.1016/j.gde.2020.05.010
Volkening A, Abbott MR, Chandra N, et al. Modeling Stripe Formation on Growing Zebrafish Tailfins. Bull Math Biol. 2020;82(5):56. Published 2020 Apr 30. doi:10.1007/s11538-020-00731-0
Author Response
We thank the reviewer for his comments on our manuscript and reference suggestions.
We now included detailed description of work on digit patterning for J. Sharpe's lab, in particular modelling-empirical crosstalk performed, mention work on zebra fish stripes, and work from the Volkening's lab, all very relevant to our discussion.